# Hot carrier cooling mechanisms in halide perovskites

Jianhui Fu[1], Qiang Xu [1], Guifang Han[2], Bo Wu [1], Cheng Hon Alfred Huan[1,3], Meng Lee Leek[1] & Tze Chien Sum [1]

Halide perovskites exhibit unique slow hot-carrier cooling properties capable of unlocking disruptive perovskite photon–electron conversion technologies (e.g., high-efficiency hot-carrier photovoltaics, photo-catalysis, and photodetectors). Presently, the origins and mechanisms of this retardation remain highly contentious (e.g., large polarons, hot-phonon bottleneck, acoustical–optical phonon upconversion etc.). Here, we investigate the fluence-dependent hot-carrier dynamics in methylammonium lead triiodide using transient absorption spectroscopy, and correlate with theoretical modeling and first-principles calculations. At moderate carrier concentrations (around $10^{18}$ cm$^{-3}$), carrier cooling is mediated by polar Fröhlich electron–phonon interactions through zone-center delayed longitudinal optical phonon emissions (i.e., with phonon lifetime $\tau_{LO}$ around $0.6 \pm 0.1$ ps) induced by the hot-phonon bottleneck. The hot-phonon effect arises from the suppression of the Klemens relaxation pathway essential for longitudinal optical phonon decay. At high carrier concentrations (around $10^{19}$ cm$^{-3}$), Auger heating further reduces the cooling rates. Our study unravels the intricate interplay between the hot-phonon bottleneck and Auger heating effects on carrier cooling, which will resolve the existing controversy.

[1] Division of Physics and Applied Physics, School of Physical and Mathematical Sciences, Nanyang Technological University, 21 Nanyang Link, Singapore 637371, Singapore. [2] Energy Research Institute @NTU (ERI@N), Research Techno Plaza, X-Frontier Block, Level 5, 50 Nanyang Drive, Singapore 637553, Singapore. [3] Institute of Materials Research and Engineering (IMRE), A*STAR, 2 Fusionopolis Way, Innovis #08-03 138634, Singapore. Correspondence and requests for materials should be addressed to T.C.S. (email: Tzechien@ntu.edu.sg)

Hot carriers refer to electrons (holes) in a Boltzmann distribution with initial kinetic energies at least $k_B T$ above the conduction (valence) bands[1]. They are formed from the subsequent thermalization of non-equilibrium photoexcited carrier populations (via carrier–carrier scattering processes within 100 fs) following the absorption of above-bandgap photons. These hot carriers (HCs) later equilibrate within several ps with the semiconductor lattice through carrier cooling processes such as carrier–phonon scattering, Auger process, etc. The mechanisms and dynamics of HC cooling in semiconductors are of fundamental importance for enhancing device functionalities (e.g., in photocatalytic activity, photodetector sensitivity, spectral range, etc.)[2–4].

Perovskite optoelectronics have recently made great strides in high performance solar cells (over 22% power conversion efficiency), high color purity light emitting diodes (over 11% external quantum efficiency), sensitive broadband detectors from visible to UV and from X-ray to gamma-ray, even tunable lasing, etc.[5–8]. These advances are driven by perovskite's novel optoelectronic properties such as low defects, strong absorption coefficient, long balanced carrier diffusion lengths, etc.[9]. Nevertheless, a deep understanding of these exceptional photophysical properties remains modest. Particularly, detailed physical insights into the origins and mechanisms of slow HC cooling in halide perovskites remain sketchy and confusing with disparate models being proposed. Clarifying them have important ramifications for light harvesting, and light emission and amplification applications[10]. For instance, slow HC cooling holds great promise for designing high-efficiency photovoltaic cells; while optical gain onsets (and thresholds) are highly dependent on the thermalization processes

at high carrier densities. Since the first report[9] of slow HC cooling in methyl ammonium lead iodide (MAPbI₃) polycrystalline thin films by Xing et al. in 2013, Li et al.[11] successfully slowed down the cooling by a further two orders with MAPbBr₃ nanocrystals and efficiently extracted their HCs. This later work epitomizes the potential of perovskites for HC photovoltaics. Presently, slow HC cooling in halide perovskite thin films has been attributed to various contributions such as large polaron screening effect[12], hot-phonon effect[13], or acoustical–optical phonon upconversion[14] that are still under intense debate. A clear understanding of these intrinsic photophysics plays a vital role in developing disruptive perovskite optoelectronic technologies.

Herein, we explicate the origins and the mechanisms of fluence-dependent slow HC cooling in the archetypal solution-processed MAPbI₃ perovskite films. Using transient absorption (TA) spectroscopy, we probe the carrier distribution function and show that HC cooling exhibits a clear dependence on the initial carrier density and excitation energy. At moderate carrier densities (around $10^{18}$ cm⁻³), slow HC cooling is dominated by polar Fröhlich interactions via zone-center non-equilibrium longitudinal optical (LO) phonon emissions (with LO phonon lifetimes of $0.6 \pm 0.1$ ps). First-principles calculations reveal a large energy separation between the LO phonon ($\hbar\omega_{LO} \approx 8$ meV) and longitudinal acoustic (LA) phonon ($\hbar\omega_{LA} = 2.5$ meV) branches in MAPbI₃ perovskite. The absence of an efficient Klemens channel for LO phonon decay gives rise to the hot-phonon bottleneck in perovskites. HC cooling is further retarded by Auger heating at high carrier densities (around $10^{19}$ cm⁻³). On the other hand, free-carrier screening has a negligible effect on HC cooling. Our findings discern the intricate interplay between the hot

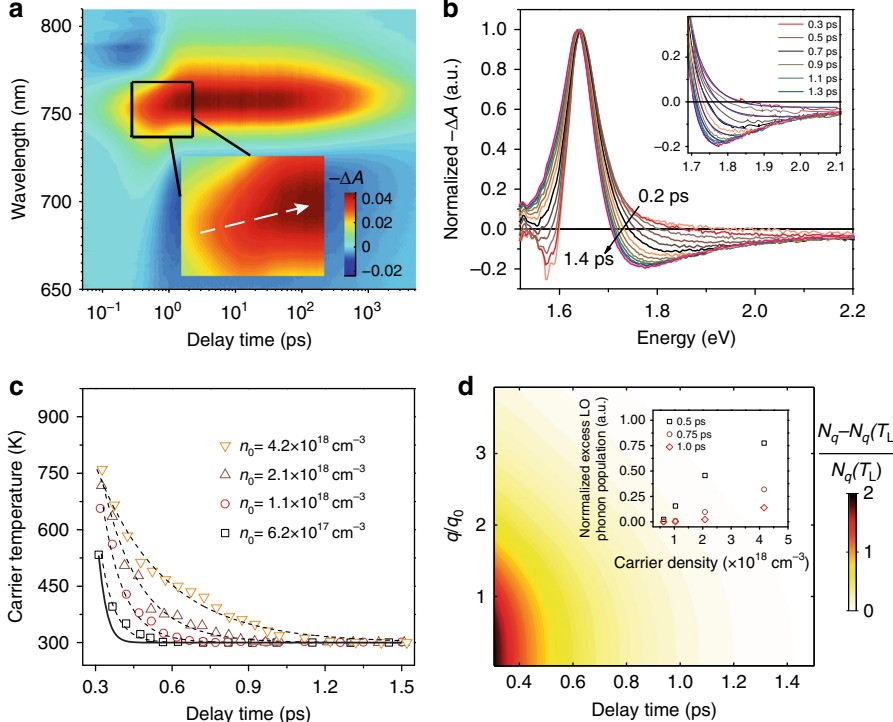

**Fig. 1** TA spectra and HC cooling dynamics. **a** Representative pseudo-color TA spectra plot of MAPbI₃ films excited at 2.48 eV with initial carrier density $n_0$ of $4.2 \times 10^{18}$ cm⁻³. The peak intensity red-shifts with increasing time delay from 0.1 to 1.5 ps (inset—white dotted arrow as a guide to the eye). **b** Normalized TA spectra extracted from **a** with variable delays from 0.2 ps to 1.4 ps and the inset shows the representative fitted high energy tails using the Maxwell–Boltzmann distribution function to extract the HC temperature. **c** Extracted HC temperature with delay at different carrier density. Black line: calculated HC cooling dynamics in the absence of hot-phonon effect. All the dashed lines are calculated HC cooling dynamics in the presence of hot-phonon effect (see main text). **d** Pseudo-color plot of calculated relative non-equilibrium hot LO phonon distribution (i.e., $[N_q - N_q(T_L)]/N_q(T_L)$) as a function of normalized phonon wave vector $q/q_0$ and delay time for $n_0$ of $4.2 \times 10^{18}$ cm⁻³. Inset shows the carrier-density-dependent hot LO phonon population with a typical phonon wave vector $q = q_0 = \sqrt{2m_e\omega_{LO}/\hbar}$ at delays of 0.5, 0.75, and 1.0 ps

phonon and Auger heating effects on carrier cooling in perovskites over a broad range of carrier densities relevant for light harvesting, light emission and amplification applications.

## Results

**Pump fluence-dependent hot-phonon effect.** Solution-processed $MAPbI_3$ polycrystalline thin films with excellent morphological and optoelectronic properties were prepared for this study. The preparation details are given in methods. Briefly, XRD (Supplementary Fig. 1), SEM, (Supplementary Fig. 2a) and step-profile measurements (Supplementary Fig. 2b) indicate excellent crystalline quality with grain size of around $80 \pm 10$ nm and a film thickness of around $60 \pm 10$ nm. The small exciton binding energy measured ($E_b$ around $6.3 \pm 0.1$ meV much smaller than $k_B T$ around 25 meV, Supplementary Fig. 3 and Supplementary Note 1) shows that free electrons and holes are the primary photoexcited species. The solution-prepared films also possess exceptionally low trap densities of around $8.9 \pm 0.6 \times 10^{16}$ cm$^{-3}$ (Supplementary Fig. 4 and Supplementary Note 2).

Figure 1a shows a pseudo-color TA plot of $MAPbI_3$ films excited at 2.48 eV with a fluence of 16 μJ cm$^{-2}$ or initial carrier density $n_0$ of $4.2 \times 10^{18}$ cm$^{-3}$ (estimated from the pump fluence and absorption coefficient—see Supplementary Note 1). The TA plot shows several obvious features: negative absorption change ($\Delta A < 0$) centered around the bandgap (1.64 eV), which is also known as the photobleaching (PB) band that arises from the band filling effect; below bandgap (smaller than 1.6 eV) photoinduced absorption (PIA$-\Delta A > 0$) at early times (before 0.6 ps) that later evolved to a PB signature; and above-bandgap PIA arising from the change in the imaginary part of the refractive index due to the injection of free carriers[15]. In addition, we also observed a small red-shift of the PB peak position (Fig. 1a, inset), that becomes even more obvious at high fluence (Supplementary Fig. 5). The bandgap of perovskite is strongly influenced by the Burstein–Moss effect and band gap renormalization (BGR), which are competing carrier-density-dependent effects (Supplementary Fig. 6 and Supplementary Note 3)[16]. At short timescales (before 2 ps), the tradeoff from BGR (causing red-shift) and Burstein–Moss effect (causing blue-shift) yields a slightly red-shifted PB peak (Fig. 1a, inset).

Figure 1b shows the normalized TA spectra of $MAPbI_3$ films (within the first 2 ps) extracted from the pseudo-color TA plot of Fig. 1a. The $\Delta A$ response is a direct correlation of the electron and hole Fermi distributions (Supplementary Note 4). At the high energy tail, the Fermi distribution function can be approximated by a Maxwell–Boltzmann distribution function. To ensure that the HCs have redistributed their energies and reached a quasi-temperature, we analyzed the HC cooling dynamics after a 0.3 ps delay. We fitted the high energy tail of the TA spectra (i.e., between 1.7 eV and 2.1 eV) using the Maxwell–Boltzmann distribution function to extract the HC temperature (Fig. 1b, inset). Both electrons and holes contribute to the TA response. Given their similar effective masses ($m_e = 0.19m_0$, $m_h = 0.25m_0$), their contributions to the HC cooling process will be roughly the same [13,17,18]. Therefore, for convenience, we assume that the extracted HC temperature to be the electron's ($T_c \approx T_e$), which will not change the conclusions here (Supplementary Note 4). Figure 1c shows the HC cooling dynamics with increasing carrier densities (or pump fluence). At higher carrier concentrations, the slope of HC cooling curve becomes gentler, indicating a lower energy loss rate. In addition, for carrier concentrations $n_0$ of $2.1 \times 10^{18}$ cm$^{-3}$ and $4.2 \times 10^{18}$ cm$^{-3}$, HCs require more than 1 ps to equilibrate with the lattice, exhibiting a slower HC cooling process at higher carrier concentrations. This observation concurs

with the slower band-edge PB rise time at higher pump fluence (Supplementary Fig. 7).

We correlate these TA spectra signatures with the underlying photophysical processes to gain deeper insights into the slow HC cooling mechanism(s). Beginning with the above band-edge photon absorption process for the direct bandgap $MAPbI_3$ polar semiconductor and neglecting the small redshift of the PB peak (discussed earlier), $\Delta A$ can therefore be approximated by:

$$\Delta A(E) = -A_0(E)[f_e(E_c) + f_h(E_v)] \qquad (1)$$

where $A_0(E)$ is linear absorbance, $f_{c,v}(E_{c,v}) = \left[1 + \exp[(\pm E \mp E_{Fc,v})/(k_B T)]\right]^{-1}$ is the Fermi–Dirac distribution (Supplementary Note 4). HC cooling is a complex process involving an interplay of carrier–carrier, carrier–phonon, and phonon–phonon interactions that is strongly influenced by the semiconductor's band structure. To simplify the carrier–phonon interaction process, we consider a direct bandgap polar semiconductor with parabolic band structure following femtosecond pulse excitation. The electrons in the valance band will be photoexcited to the conduction band with excess energies $\Delta E_i = \frac{m_r}{m_i}(\hbar\omega - E_g)$, where $m_r$ is the reduced mass, $m_0 em_i$ is the $i$ ($i = $ e, h) carrier mass, and $\hbar\omega$ is the photon energy. During carrier thermalization (occurring within 100 fs), the energetic carriers will rapidly redistribute their energies via elastic carrier–carrier, carrier–ion collisions, and intervalley scattering and thermalize to a quasi-equilibrium state that can be characterized by a quasi-temperature $T_c$[19].

Concurrent with carrier energy redistribution among the HCs is the modification of their Fermi distribution function. HCs will lose their excess energy through inelastic electron–phonon coupling interactions. There are generally two coupling mechanisms that drive electron–phonon scattering in inorganic semiconductors: polar Fröhlich interactions of electron–LO phonon scattering for polar crystals that arises from Coulomb interactions between the electrons and the macroscopic electric field induced by the out-of-phase displacements of oppositely charged atoms caused by the LO phonon mode[20]; and electron-acoustic phonon scattering that is efficient at low temperature (below 30 K) via non-polar deformation potential interactions and piezoelectric interactions whose electronic structure is modified by the lattice distortion and the induced macroscopic polarization field, respectively. At room temperature, polar Fröhlich interactions via LO phonon emission is the dominant relaxation pathway for hot electrons in polar semiconductors. Such LO phonon modes will decay into daughter phonon modes via phonon–phonon interactions until the electron–LO phonon system equilibrates with the lattice's thermal reservoir. If the time scale for relaxation of emitted LO phonons into the acoustic phonons' heat bath is comparable or longer than the characteristic HC relaxation time, then an appreciable non-equilibrium population of LO phonons exists, which leads to reduced LO phonon emission that will slow the HC cooling accordingly[21]. This is the well-known hot-phonon effect that is commonly observed in highly excited polar semiconductors such as CdS[22], CdSe[23], GaAs[23], and GaN[24]. Apart from the hot-phonon effect, another possible origin of slow HC cooling is the Coulomb screening effect that stems from the shielding of the carrier–phonon scattering by the high density carrier plasma. However, free-carrier screening effect on HC cooling is indeed negligible for $MAPbI_3$, which is corroborated by our own findings (see later discussion).

From Fermi's golden rule, the rate of increase of a phonon population via LO phonon emission and absorption (in the absence of screening effect) for carrier concentrations $n_0$ below

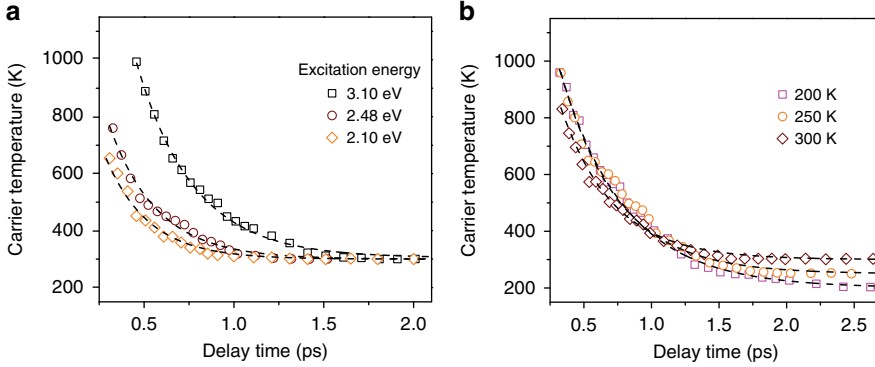

**Fig. 2** HC cooling governed by the hot-phonon bottleneck effect at carrier densities around $10^{18}$ cm$^{-3}$. **a** HC cooling of MAPbI$_3$ films excited with same $n_0$ of $4.2 \times 10^{18}$ cm$^{-3}$ but varying excitation energy. **b** Temperature-dependent HC cooling dynamics of MAPbI$_3$ films excited at 2.48 eV with initial carrier density $n_0$ of $5.5 \times 10^{18}$ cm$^{-3}$. The dashed lines are calculated HC cooling dynamics using our model (see main text)

$10^{19}$ cm$^{-3}$ is described by the following expression[25,26]:

$$\left(\frac{\partial N_q}{\partial t}\right)_{e-ph} = \frac{m^2 k_B T_c}{\pi \hbar^5 q} |M_q|^2 (N_q(T_c) - N_q)$$

$$\times \ln\left\{\frac{1 + \exp\left[\eta - \frac{\hbar^2}{8mk_B T_c}\left(q - \frac{2m\hbar\omega_q}{\hbar^2 q}\right)^2\right]}{1 + \exp\left[\eta - \frac{\hbar^2}{8mk_B T_c}\left(q + \frac{2m\hbar\omega_q}{\hbar^2 q}\right)^2\right]}\right\} \quad (2)$$

where $\eta = E_f/k_B T_c$ is the quasi-Fermi level, $N_q(T_c) = 1/(\exp(\hbar\omega_q/k_B T_c) - 1)$ is the Bose distribution of LO phonons at HC temperature $T_c$. $|M_q|^2 = \frac{2\pi\hbar^2 eE_0}{mq^2}$ is the transition matrix element; and for the expression $eE_0 = \frac{me^2\hbar\omega_{LO}}{4\pi\varepsilon_0\hbar^2}\left[\frac{1}{\varepsilon_{Opt}} - \frac{1}{\varepsilon_{Stat}}\right]$, $m$ is the electron effective mass, $\varepsilon_{Opt}$ and $\varepsilon_{Stat}$ are respectively the optical and static dielectric constants, $\varepsilon_0$ is the vacuum permittivity, $\hbar\omega_q$ is the LO phonon energy, and $e$ is the electric charge unit. Given that the transition matrix $|M_q|^2$ scales inversely with $q^2$, electron–phonon scattering arises from the LO phonons near the zone-center with small $q$.

Next, the decay of LO phonons into daughter phonon modes using a single mode relaxation time approximation can be described as follows:

$$\left(\frac{\partial N_q}{\partial t}\right)_{ph-ph} = -\frac{N_q - N_q(T_L)}{\tau_{ph}} \quad (3)$$

where $\tau_{ph}$ is the LO phonon lifetime, $N_q(T_L) = 1/(\exp(\hbar\omega_q/k_B T_L) - 1)$ is the equilibrium LO phonon occupation number at the lattice temperature $T_L$. The LO phonon lifetime $\tau_{ph}$, which is associated with the anharmonic interactions between LO phonon and acoustic phonons, is in general temperature dependent and varies for different phonon–phonon coupling interactions[27]. There are typically three decay routes for LO phonon thermalization: the Klemens channel (i.e., decay into two counter-propagating LA phonons[28]); the Ridley channel (i.e., decay into a transverse optical (TO) phonon and an acoustic phonon (TA or LA)[29]); and the Barman–Srivastava channel (i.e., decay into two TO phonons)[30]. Amongst them, the Klemens channel is most efficient for LO phonon relaxation as the daughter LA phonons can quickly thermalize with the lattice[27]. However, there have been no reports on direct experimental measurements of $\tau_{ph}$ in lead halide perovskites. Calculation of $\tau_{ph}$ demands detailed analysis of anharmonic phonon–phonon scattering, which is beyond the scope of our present study. Therefore, we simply assume $\tau_{ph}$ is a parameter that only depends on the lattice temperature, which is a reasonable assumption.

Here, we posit that the hot-phonon bottleneck in perovskites arises from an inefficient Klemens channel for LO phonon decay.

From Klemens theory[28], a semi-empirical expression for the temperature dependent $\tau_{ph}$, which is associated with the decay of LO phonon into two daughter acoustic phonons with energy of $\frac{1}{2}\hbar\omega_{LO}$ each is:

$$\tau_{ph}(T_L) = \tau_{ph}(0)\left[1 + 2\left[\exp\left(\frac{\hbar\omega_{LO}}{2k_B T_L}\right) - 1\right]^{-1}\right]^{-1} \quad (4)$$

where $\tau_{ph}(0)$ is the LO phonon lifetime at $T_L = 0$ K. Both momentum and energy are conserved in the relaxation of the LO phonon into the daughter acoustic phonons.

To simplify the electron–LO phonon interactions, we assume that the LO phonons follow a spherical distribution and are dispersionless in the Brillouin zone. From Eq. (2), we obtained the energy loss rate of the HC:

$$\left\langle\frac{dE}{dt}\right\rangle = -\frac{m^2 k_B T_c}{2\pi^3 \hbar^5 n}\int_0^\infty dq |M_q|^2 \hbar\omega_{LO}(N_q(T_c) - N_q)$$

$$\times \ln\left\{\frac{1 + \exp\left[\eta - \frac{\hbar^2}{8mk_B T_c}\left(q - \frac{2m\hbar\omega_{LO}}{\hbar^2 q}\right)^2\right]}{1 + \exp\left[\eta - \frac{\hbar^2}{8mk_B T_c}\left(q + \frac{2m\hbar\omega_{LO}}{\hbar^2 q}\right)^2\right]}\right\} \quad (5)$$

where $n$ is the initial carrier density and $N_q$ is the non-equilibrium LO phonon occupation number. In the absence of hot-phonon effect at low carrier concentrations, $N_q \approx N_q(T_L)$, a simplified expression of Eq. (5) can be obtained:

$$\left\langle\frac{dE}{dt}\right\rangle = -P_0\frac{\exp(-x_c) - \exp(-x_L)}{1 - \exp(-x_L)}\left[\frac{(x_c/2)^{1/2}e^{x_c/2}K_0(x_c/2)}{\sqrt{\pi/2}}\right], \quad (6)$$

where $x = \frac{\hbar\omega_q}{k_B T}$, $K_0(x)$ is the zero order Bessel function, and $P_0 = eE_0\left(\frac{2\hbar\omega_{LO}}{m}\right)^{\frac{1}{2}}$ is the effective energy loss rate. For MAPbI$_3$ perovskite[17,20,31], $\hbar\omega_{LO} \approx 13$ meV, $m_e = 0.19\,m_0$, $\varepsilon_{Opt} = 6.5$, $\varepsilon_{stat} = 70$, the effective scattering rate can be calculated: $W_0 = \frac{P_0}{\hbar\omega_{LO}} \approx 2.7 \times 10^{13}$ s$^{-1}$.

Figure 1c shows the global-fitted fluence-dependent HC cooling dynamics (dashed lines) using Eq. (5). In the absence of any hot-phonon effect, HCs rapidly lose their excess energies due to an extremely high electron–LO phonon scattering rate (solid line—Eq. (6)). This scenario deviates from our experimental data. Hence, slow HC cooling in MAPbI$_3$ is a distinct signature of the hot-phonon effect. We globally fitted the HC cooling dynamics

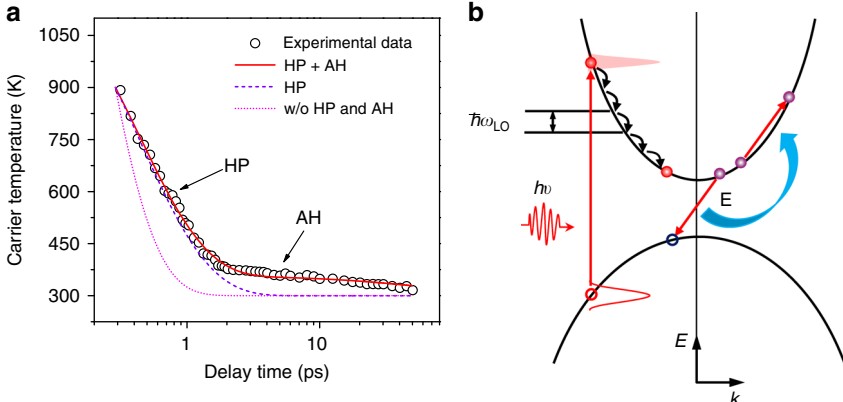

**Fig. 3** HC cooling governed by hot-phonon bottleneck and Auger heating effects at carrier densities above $10^{19}\,\text{cm}^{-3}$. **a** HC cooling dynamics following photoexcitation at 2.48 eV with a carrier density $n_0$ of $10.4 \times 10^{18}\,\text{cm}^{-3}$ at RT. Black circles: HC temperature extracted from TA spectra. The lines show the calculated HC cooling dynamics for $\tau_{\text{ph}} = 0.6$ ps: with hot-phonon (HP) effect only (violet dashed line); with both HP and Auger heating (AH) effects (bright red line); and without HP and AH effects (magenta dotted line). **b** Schematic of hot electron relaxation process via LO phonon emission and non-radiative Auger recombination that contributes to further deceleration of hot electron cooling. The same process will also be present for the hot holes (not shown for clarity)

for four different carrier densities and obtained a LO phonon lifetime $\tau_{\text{ph}} = 0.6 \pm 0.1$ ps, which qualitatively agrees with a recent experimental result of $\tau_{\text{ph}} \approx 0.1$ ps in the same order[32]. Figure 1d shows a pseudo-color plot of calculated relative nonequilibrium hot LO phonon population as a function of normalized phonon wave vector ($q/q_0$) and delay time for $n_0$ of $4.2 \times 10^{18}\,\text{cm}^{-3}$. Obviously, HC cooling follows the relaxation of hot LO phonon and is driven predominantly by the LO phonons with smaller wave vectors. As a self-consistency check, we also calculated the relative excess LO phonon populations (i.e., $[N_q - N_q(T_L)]/N_q(T_L)$) with a typical phonon wave vector of $q = q_0 = \sqrt{2m_e\omega_{\text{LO}}/\hbar}$ at three different delay times of 0.5 ps, 0.75 ps, and 1.0 ps (Fig. 1d, inset). The non-equilibrium LO phonon population increases with increasing carrier densities. The slower HC cooling at higher carrier concentrations (Fig. 1c) is therefore consistent with the growth of the non-equilibrium LO phonon population. Next, excitation energy and temperature-dependent HC cooling dynamics were also investigated as further tests of our model.

**Excitation energy and temperature-dependent hot-phonon effect.** Figure 2a shows the HC cooling dynamics of MAPbI$_3$ films excited at different energies but with the same carrier density $n_0$ of $4.2 \times 10^{18}\,\text{cm}^{-3}$. The experimental data can be well-fitted using Eq. (5) with $\tau_{\text{ph}} = 0.6 \pm 0.1$ ps. This suggests that at carrier densities around $10^{18}\,\text{cm}^{-3}$, the hot-phonon effect is the dominant mechanism governing the slow HC cooling and is independent of excitation energy. Furthermore, we observed a slower rise of the band-edge PB response in the short time range for higher energy excitation (Supplementary Fig. 8a) at the same carrier concentration. This is because the excess energy gained by the electrons from even higher energy excitation will lead to larger population build-up of non-equilibrium LO phonons (Supplementary Fig. 8b). Hence, HCs take more time to cool down to the lattice thermal reservoir, in agreement with our model. Figure 2b shows the temperature-dependent HC cooling dynamics of MAPbI$_3$ films excited at 2.48 eV with an initial carrier concentration of $5.5 \times 10^{18}\,\text{cm}^{-3}$. We restrict to temperatures above MAPbI$_3$'s orthorhombic to tetragonal phase transition (around 160 K)[20] to minimize the effects of temperature-dependent band structure variation that would complicate our analysis. As expected, HCs require more time to cool down at lower lattice temperatures, where the fitted LO phonon lifetimes at 200 K, 250 K, and 300 K are: $\tau_{\text{ph}}(200\,\text{K}) = 1.5 \pm 0.1$ ps, $\tau_{\text{ph}}(250\,\text{K})$

$= 1.1 \pm 0.1$ ps, $\tau_{\text{ph}}(300\,\text{K}) = 0.6 \pm 0.1$ ps, respectively. The increase of $\tau_{\text{ph}}$ with decreasing temperature arises from a decrease of phonon–phonon scattering rates due to the associated decrease in phonon occupancy[33]. This leads to larger non-equilibrium LO phonon populations (Supplementary Fig. 9). Consequently, slower HC cooling is observed at lower temperatures, again in concurrence with our model.

**Interplay of hot phonon and Auger heating effects.** We next focus on the HC cooling process at even higher carrier concentrations above $10^{19}\,\text{cm}^{-3}$ that are relevant for applications like optical amplification. Figure 3a shows the evolution of the HC temperatures following 2.48 eV photoexcitation with an initial carrier density of $10.4 \times 10^{18}\,\text{cm}^{-3}$ at room temperature. The data shows two distinct gradients, indicating the presence of two different HC cooling mechanisms at higher carrier concentrations. The HCs also remain at higher temperatures than the lattice temperature for a longer duration (i.e., ∼50 ps) because of the larger non-equilibrium LO phonon population. Given the good linear fit of the band-edge differential absorption ($-\Delta A$) over the broad range of carrier densities (Supplementary Fig. 10), we assume that the HCs are still non-degenerate for carrier concentrations above $10^{19}\,\text{cm}^{-3}$. The hot-phonon effect alone does not account for the data (dashed violet line in Fig. 3a). We posit that this second slow HC cooling mechanism that appears at higher carrier concentrations above $10^{19}\,\text{cm}^{-3}$ is the non-radiative Auger heating arising from the enhanced carrier–carrier interactions. This deduction is supported by the observations of rapid shortening of the band-edge PB kinetics that is typical for multi-particle Auger recombination process (Supplementary Fig. 5) and persistent higher energy tail of hot PL (Supplementary Fig. 11). There are generally two distinct types of Auger processes: the intraband Auger recombination and the interband Auger recombination (or Auger heating). The latter process (present here) involves the non-radiative transfer of the e–h recombination energy to another electron (or hole) that results in the excitation of this carrier to even higher energies (Fig. 3b)[34]. The recombination energy transferred to the electronic system is proportional to $E_g + E$, where $E_g$ is the band gap[35]. Although e–h recombination energy transfer does not change the total energy of the e–h pair, the heating of the electronic system will still slow down HC cooling.

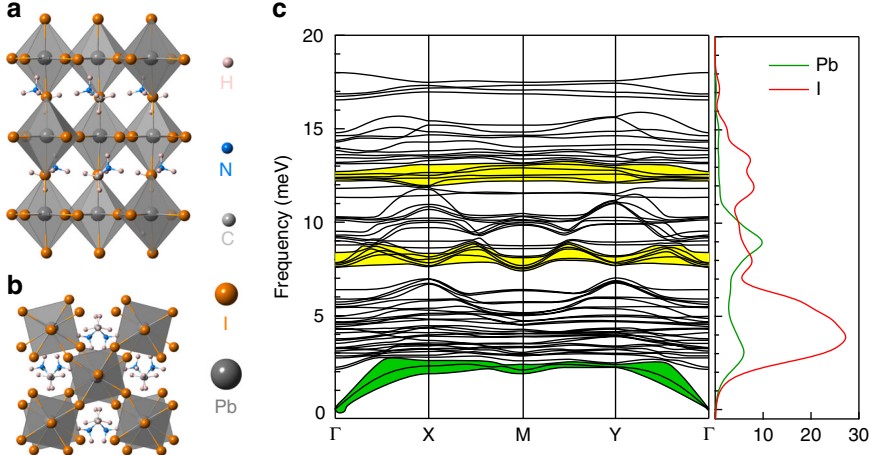

**Fig. 4** The optimized crystal structures and phonon dispersion of the low frequency modes of tetragonal MAPbI$_3$. **a** Side view, **b** top view. **c** Phonon dispersion spectra and the corresponding projected density of states. The two yellow zones are the Pb–I stretch vibrations, which are centered around 8 and 12 meV, respectively. The green zone is the acoustic phonon mode with maximum frequency around 2.5 meV. Details of the calculations are given in Supplementary Note 9

Accounting for the Auger heating effect in HC cooling, the total energy loss rate can be described by the following kinetic equations:

$$\left\langle \frac{\mathrm{d}E}{\mathrm{d}t} \right\rangle_{\mathrm{tot}} = \left\langle \frac{\mathrm{d}E}{\mathrm{d}t} \right\rangle_{\mathrm{e-ph}} + k_3 n^2 (E_{\mathrm{g}} + E) \quad (7)$$

$$\frac{\mathrm{d}n}{\mathrm{d}t} = -k_1 n - k_2 n^2 - k_3 n^3 \quad (8)$$

The first term on the right side of Eq. (7) refers to the energy loss rate from electron–LO phonon scattering; while the second term represents the contribution from Auger heating, where $E_{\mathrm{g}}$ is the bandgap and $n$ is the carrier concentration whose rate of change is in turn dependent on the various recombination orders (Eq. (8)), where $k_1$ is the monomolecular recombination rate via defects, $k_2$ is the free-carrier bimolecular recombination rate, and $k_3$ is the three body non-radiative Auger recombination rate.

Given that the band-edge PB amplitude scales linearly with carrier concentration, we estimate the recombination rates by global fitting the band-edge PB kinetics with varying fluence (Supplementary Figs. 10 and 12 and Supplementary Note 5). The rates are: $k_1 = 2.3 \pm 0.1 \times 10^7 \mathrm{\ s^{-1}}$, $k_2 = 2.5 \pm 0.2 \times 10^{-11} \mathrm{\ cm^3 \ s^{-1}}$, and $k_3 = 5.4 \pm 0.1 \times 10^{-29} \mathrm{\ cm^6 \ s^{-1}}$, respectively; which are consistent with literature values[36]. Our modified slow HC cooling model (that includes both hot phonon and Auger heating effects), yields a reasonable fit of the experimental data (solid red line in Fig. 3a). For comparison, HC cooling in the absence of both hot phonon and Auger heating effects (magenta dotted line); and in the absence of Auger heating effect only (violet dashed line) are also plotted in Fig. 3a to illustrate the deviation from the experimental data. Our model clearly shows that at high carrier densities, the non-equilibrium LO phonon population dominates the HC cooling in the first 2 ps, while Auger heating contributes to further retardation of HC relaxation. This extends the slow HC cooling to over 50 ps in lead iodide perovskites. At these carrier concentrations (above $10^{19} \mathrm{\ cm^{-3}}$), the free-carrier screening effect has negligible effect on HC cooling (Supplementary Note 6 and Supplementary Fig. 13).

## Discussion

The similarity of perovskite's electron and hole effective masses aptly highlights its distinct advantage over conventional semiconductors for HC applications. In typical semiconductors, their electron effective mass is much lower than that of the hole. This will result in faster hot hole cooling compared to hot electron cooling. In contrast, the more "balanced" slow hot electron and hot hole cooling in perovskites will be more amenable for developing practical hot-carrier optoelectronic devices. Possible origins of slow HC cooling in semiconductors include: hot-phonon effect, Coulomb screening effect, and Auger heating effect.

Firstly, regarding the hot-phonon effect, there is a complex interplay of two factors: the carrier density and the LO phonon lifetime. The presence of a hot-phonon bottleneck is not simply determined by the absolute phonon lifetime $\tau_{\mathrm{ph}}$, but by the competition between the phonon decay rate ($k_{\mathrm{ph}} = 1/\tau_{\mathrm{ph}}$) and the electron–phonon scattering rate ($k_{\mathrm{e-p}}$ which is proportional to the energy loss rate), that are in turn dependent on the carrier density[37]. In general, under low excitation conditions ($n$ below $10^{17} \mathrm{\ cm^{-3}}$), the hot-phonon effect is negligible and the energy loss rate of the HC is independent of carrier density. However, under highly excited conditions with carrier density well above $10^{18} \mathrm{\ cm^{-3}}$, significant slow HC cooling can be observed due to the presence of a large non-equilibrium LO phonon population, which leads to reduced net LO phonon emission. This highly excited regime is characterized by an obvious reduced energy loss rate with carrier density, i.e., slower HC cooling. For the second factor, long LO phonon lifetime can enhance the hot-phonon effect by lengthening the LO phonon decay given that HC cooling is a cascade process between electron–LO phonon and anharmonic phonon–phonon interactions. For lead halide perovskites, the hot-phonon effect is nontrivial even at moderate carrier densities (Supplementary Table 1). Our model yields a fitted LO phonon lifetime of around $0.6 \pm 0.1$ ps at room temperature, which is much longer than typical electron–LO phonon scattering time constant of approximately 13 fs (Supplementary Note 7). We performed first-principles calculations in a bid to uncover the puzzle of this long LO phonon lifetime. Figure 4 shows the optimized (a) side view, and (b) top view of the unit cell and (c) the phonon dispersion spectra and corresponding projected density of states for the low frequency modes of a tetragonal MAPbI$_3$ with phase group of $I4/mcm$ ($D_{4h}^{18}$). More details of the calculated phonon spectra and phonon modes are given in Supplementarys Fig. 14 and 15. From Fig. 4c, the Pb–I stretch vibration (LO) frequencies are centered

around 8 meV and 12 meV (i.e., 65 and 99 cm$^{-1}$, two yellow zones), while the maximum acoustic phonon frequency lies around 2.5 meV (i.e., 20 cm$^{-1}$, green zone). Our Raman measurements and DFT calculations (Supplementary Fig. 16) are in good agreement with literature reports[32,38,39]. The much larger LO phonon energies compared to their LA energies suggest a large energy separation ($E_{LO} > 2E_{LA}$) between the LO and LA phonon branches. The efficient Klemens channel for LO phonon decay is thus hindered and consequently, the LO phonons could only decay via alternative less efficient channels, such as the Ridley channel. The inhibition of Klemens channel can be enhanced when there is a large atomic mass difference between the positively and negatively charged ions. For instance, a longer LO phonon lifetime was observed for InP (around 7.6 ps) in contrast to GaAs (around 2.1 ps) at room temperature[40]. The relatively small LO phonon energy (around 8 meV) for MAPbI$_3$ is nonetheless expected, considering that the zone-center LO phonon mode mainly stems from the out-of-phase stretching of Pb$^{2+}$ and I$^-$ in heavy inorganic PbI$_6$ cage[41]. This suggests that more LO phonons are needed to cool down the HCs with the same excess energy; which inevitably leads to a large non-equilibrium LO phonon population. Furthermore, the low density of states (DOS) of the electronic structure in MAPbI$_3$ could also enhance the hot-phonon effect[42]. This is because a small DOS will lead to a reduction of the available relaxation pathways as well as small effective mass, which can aggravate the hot-phonon effect by reducing the energy loss rate of the HC (Supplementary Fig. 17 and Supplementary Note 8). Hence, the suppression of the efficient Klemens channel for LO phonon decay, together with the large non-equilibrium LO phonon population and small DOS of the electronic structure, give rise to the strong hot-phonon bottleneck effect in lead halide perovskites. Note that the LO phonon mode in perovskites is dominated by the stretching vibration of the Pb–I networks and similar pump fluence-dependent slow HC cooling phenomenon are also observed in CsPbI$_3$ films (Supplementary Fig. 18); which suggests that the organic cation may not play a significant role in the slow HC cooling. More detailed theoretical and experimental studies are needed to investigate the other relaxation channels of LO phonons and their corresponding effects on HC cooling.

Secondly, for the Coulomb screening effect, HC cooling process could also be retarded because the macroscopic electric field induced by the out-of-phase displacements of the atoms is weakened by the Coulomb screening. The free-carrier Coulomb screening effect is in general carrier-density dependent and varies for different materials[43]. In general, for various semiconductor systems, the screening effect at carrier concentrations below 10$^{19}$ cm$^{-3}$ plays a negligible role in the retardation of carrier cooling (unlike the dominant hot-phonon effect)[23]. This may be because at low carrier densities, the polarization waves of the system are still the polar optical phonons of the lattice, therefore having little effect on the static dielectric constants in these systems. However, under intense excitations (above 10$^{19}$ cm$^{-3}$), enhanced shielding effects from high carrier density plasma may still have a significant influence on the slow HC cooling as the polarization waves are being modified and this changes the dielectric constants. In the case of halide perovskites, our findings show that the screening effect has insignificant influence on the slow HC cooling at low pump fluence (Supplementary Fig. 13). Furthermore, the screening effect under intense excitations also has negligible influence on the HC cooling. This is possibly due to perovskite's large static dielectric constant ($\varepsilon_S \sim 70$—arising mainly from the vibration modes of the inorganic PbI$_3$ network) that creates a large screening length (hence, weaker screening).

Lastly, we discuss the contributions of Auger heating on slow HC cooling. Although Auger heating is typically insignificant in

bulk semiconductors, it can still play a dominant role in slow HC cooling at high excitation densities due to the stronger carrier–carrier interactions. According to Fermi's golden rule[44] $C \propto \sqrt{E_g} \exp\left[-\frac{m_e}{m_e + m_h}\frac{E_g}{k_B T}\right]$, where $m_h$ is the effective mass of the hole, the Auger recombination rate is band gap and carrier temperature dependent. The strong Auger heating in lead iodide perovskites at high excitation densities is due to its relatively smaller band gap and the dominant hot-phonon effect. One could envisage that Auger heating would have a more significant effect on HC cooling in perovskite nanostructures (as in the case for inorganic semiconductor nanostructure systems such as CdSe quantum rods[34] and GaAs/AlAs multiple quantum wells[45]) due to momentum conservation and larger overlap of the carrier wavefunctions under quantum confinement. Indeed, this has been recently reported for MAPbBr$_3$ nanocrystals[11]. Nonetheless, one should note that while Auger heating slows down HC cooling, it also simultaneously depletes the photoexcited carrier population. This is an important consideration for developing practical hot-carrier devices.

In retrospect, we explicated the fluence-dependent HC cooling mechanisms in MAPbI$_3$ perovskite. At moderate carrier concentrations (around 10$^{18}$ cm$^{-3}$), slow HC cooling is primarily mediated by the hot-phonon effect with a LO phonon lifetime of $0.6 \pm 0.1$ ps. At high carrier concentrations (above 10$^{19}$ cm$^{-3}$), an interplay of both hot phonon and Auger heating effects further retard the HC cooling. Moreover, slower HC cooling with decreasing temperature can be attributed to the decrease of LO phonon occupation number. The relatively long LO phonon lifetime of $0.6 \pm 0.1$ ps for MAPbI$_3$ at room temperature is attributed to the suppression of Klemens channels for LO phonon decay. Our findings support the suppression of the Klemens channel and the Auger heating process as the origins to the slow HC cooling effect in perovskites. This contrasts with a recent report[14] proposing the acoustic–optical phonon up-conversion mechanism[46] to be responsible for the phonon bottleneck effect. Their underlying assumption[14] was based on three-dimensional and two-dimensional halide perovskites possessing similar thermal properties, which need not be the case[47]. Our present study provides fresh insights into the interplay of the hot-phonon effect and Auger heating effect on slow HC cooling in lead iodide perovskites, which are essential for a clear understanding of novel photophysics and the development of new perovskite-based optoelectronics.

## Methods

**Sample fabrication.** The MAPbI$_3$ precursor solution was prepared by dissolving 1:1 molar stoichiometric ratios of CH$_3$NH$_3$I (Dyesol) and lead iodide (TCI, 99.99%, trace metals basis) in N,N-dimethylformamide (DMF, Sigma Aldrich anhydrous, ≥99%) with 10 wt% concentration. The films were prepared by spin coating the precursor on quartz substrates at 5,000 r.p.m. for 12 s and anti-solvent of Diethyl ether (Sigma Aldrich, anhydrous, ≥99%) was dropped 3 s after the start of spin coating, followed by thermal annealing at 100 °C for 30 min. All the preparation was conducted inside a nitrogen filled glove box.

**Characterization.** XRD patterns were collected using a Bruker-AXS D8 Advance X-ray diffractometer equipped with Cu K$\alpha$ ($\lambda = 1.5418$ Å) X-ray source. The measurements were performed in 2$\theta$ mode between 10° and 50° with step size of 0.02° and the integration time of 1 s per step. Scanning electron microscopy (SEM) measurements were performed using a JEOL JSM6700F field emission scanning electron microscope operated at 5 kV. Linear absorption, optical transmittance, and total reflectance spectra of CH$_3$NH$_3$PbI$_3$ films on quartz substrate and blank quartz substrate were measured with a UV–VIS–NIR spectrophotometer (Shimadzu UV-3600) equipped with an integrating sphere (ISR-3100).

**Transient absorption spectroscopy.** Transient absorption spectra were collected using an integrated Helios and EOS setup (Ultrafast Systems LLC) operating in a nondegenerate pump—probe configuration. The pump pulses were generated from a Coherent TOPAS-C optical parametric amplifier pumped by a 1 kHz regenerative

amplifier (Coherent Legend, 800 nm, 150 fs). The amplifier was seeded by a mode-locked Ti-sapphire oscillator (Coherent Vitesse, 100 fs, 80 MHz). 400 nm pulse pump was generated by doubling 800 nm pulse with a beta barium borate (BBO) crystal. The white light continuum probe pulses were produced by focusing the 800 nm fs pulses through a 2 mm-sapphire plate. A 750 nm short pass filter was placed before the sample to filter out the 800 nm residue, that would otherwise cause strong photo-excitation and result in wrong estimate of the carrier concentration and temperature (Supplementary Fig. 19). A reference probe light before the sample and the signal probe light were collected using the same CMOS sensors.

**Time-integrated photoluminescence and time-resolved photoluminescence.**
Time-integrated photoluminescence measurement was conducted by directing the excitation laser pulses to thin films. The photoluminescence was measured at a backscattering angle of 145° by a pair of two lenses via an optical fiber coupled by a spectrometer (Acton, Spectra Pro 2500i) and CCD (Princeton Instruments, Pixis 400B). Time-resolved photoluminescence was collected using an Optronis Optoscope streak camera system with an ultimate temporal resolution of 10 ps.

**First-principles calculations.** The first-principles calculations were performed via employing the all-electron-like projector augmented wave method and the Perdew–Burke–Ernserhof revised for solids (PBEsol) exchange correlation potential as implemented in the VASP code. Structure optimization calculations were conducted by using the $5 \times 5 \times 3$ Monkhorst–Pack grid of k-points for Brillouin zone integration of MAPbI$_3$. The semicore of Pb atoms (5d orbital) are treated as valence electrons, i.e., 14 valence electrons for Pb ($5d^{10}6s^26p^2$) atom. The optimized unit cell of a tetragonal MAPbI$_3$ with space group of $I4/mcm$ ($D_{4h}^{18}$) is 8.721 Å × 8.694 Å × 12.837 Å. The cut-off energy for the plane wave expansion of the wave functions was 800 eV and the Hellman–Feynman forces were less than of 0.002 eV/Å. The phonon spectrum within the harmonic approximation were calculated using the PHONONPY code. In phonon calculations, we had used a $2 \times 2 \times 1$ supercell to ensure that the boundary atoms were not affected by the effects of the atomic displacement. The force constants were constructed based on the density functional perturbation theory approaches. More details are provided in Supplementary Note 9.

**Data availability.** The data that support the findings of this study are available from the corresponding author upon request.

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

## Acknowledgements

Financial support from Nanyang Technological University start-up grants M4080514 and M4080474; the Ministry of Education Academic Research Fund Tier 1 grants RG101/15 and RG173/16, and Tier 2 grants MOE2014-T2-1-044, MOE2015-T2-2-015 and MOE2016-T2-1-034; and from the Singapore National Research Foundation through the Singapore—Berkeley Research Initiative for Sustainable Energy (SinBeRISE) CREATE Program and the Competitive Research Program NRF-CRP14-2014-03 is gratefully acknowledged.

## Author contributions

T.C.S. and J.F. conceived the idea for the manuscript and designed the experiments. J.F. conducted the spectroscopic characterization. G.H. prepared the samples and performed the XRD and SEM measurements. Q.X. performed the first-principles calculations. M.L.L. and B.W. performed Auger rate estimation and calculation. J.F., Q.X. and T.C.S. analyzed the data and wrote the manuscript. The authors acknowledged the fruitful discussions with Prof. Subodh Gautam Mhaisalkar. All authors discussed the results and commented on the manuscript at all stages. T.C.S. led the project.

## Additional information

**Competing interests:** The authors declare no competing financial interests.

**Change history:** A correction to this article has been published and is linked from the HTML version of this paper.

