## [Peer Review File · Nature Communications]

Reviewers' comments:

Reviewer #1 (Remarks to the Author):

This manuscript reports the hot-carrier cooling mechanism of MAPbI₃ by analysis of transient absorption spectroscopy in two different ranges of carrier concentration. At moderate hot-carrier concentration, the hot carrier temperature is well fitted with the model of electron-phonon interaction including hot phonon bottleneck effect. At higher concentration, Auger heating is also prominent and the lifetime of hot carrier becomes longer. This paper is very useful for the development of photovoltaics with higher performance. However, I'd like to ask the authors some further discussion before accepted.

I have one question about LO phonon lifetime of MAPbI₃. This manuscript reported the lifetime is about 0.6 ps at room temperature. It is argued that this value is longer than the typical value of "electron-phonon" lifetime ~13 fs and hence it is the origin of hot phonon bottleneck. However, this phonon lifetime value is not quite longer than ones of other semiconductors. For example, as mentioned in the manuscript, InP and GaAs have longer lifetimes than 2 ps. Ref 30 in this article (PCCP 18 27051(2016)) describes that "the short phonon lifetimes in MAPbX₃ imply a high degree of anharmonic phonon-phonon coupling". This statement seems to be opposed to the author's statement that "The efficient Klemens channel for LO phonon decay is thus hindered" in line 325. In conclusion, I'd like to ask you some more detailed discussion about the mechanism of hot phonon bottleneck. As described in line 330-334, the large LO phonon population coming from the small LO phonon energy is plausible. In addition, the authors may be able to mention the electronic structure of MAPbI₃. For example, some papers argue that the low density of states of valence and conduction bands are related to the long electron-phonon lifetime and the high open-circuit voltage. For example, please check two papers below.

- Nano Lett, 15, 3103 (2015)

- J. Phys. Chem. C, 121, 1455 (2017)

The density state must be related to the effective mass in your model. How is the impact of the light mass on the hot phonon bottleneck?

I believe that the manuscript must be much more useful if the authors suggest parameters to differentiate MAPbI₃ from other semiconductors.

Reviewer #2 (Remarks to the Author):

Recently, slowed cooling of hot carrier has been extensively observed, and some mechanisms for hot phonon bottleneck have been proposed, including screening effect and acoustical-optical phonon upconversion. Given the significantly potential applications of hot carriers, such as high efficiency hot carrier solar cells, and photo-catalysis, a clear mechanism of the hot carrier in perovskite bulk and nanostructures are crucially important. In this study, authors experimentally observed fluence-dependent hot-carrier dynamics in perovskite film by transient absorption and time-resolved PL. They correlated the hot carrier effect with the theoretical modelling and first principles calculations. They proposed at low carrier density the hot carrier cooling is slowed down dominantly by the hot phonon bottleneck due to suppressed Klemens relaxation pathway (large energy separation between the LO and LA phonon branches), so observed cooling time of $\sim 0.6 \pm 0.1$ ps. At a high carrier density, Auger heating reduces the cooling rate; up to 50 ps cooling time was observed. Basically, the manuscript is significant, well organized and clearly presented. Authors have presented significant experimental and theoretical evidences to support their conclusions. However, I do not think really convinced, as some examples in the following. I cannot recommend accepting for publication until a significantly revision

and improvement.

1. In terms of the slowed cooling of ~ 600 fs, it is roughly in the range of a normal cooling time without any hot phonon bottleneck for most semiconductors (experimental observations). Authors compared and claimed delayed slowed cooling time based on calculated 15 fs of cooling time, for the case without hot phonon effect. It is not valid. This cooling time of 15 fs is also inconsistent with most observations of perovskites.
2. MAPbI₃ has been proposed a second valence band, this will affect the initial relaxation and thus the hot carrier relaxation, author should also consider the possible influence;
3. This is well-known that hot phonon effect that is commonly observed in highly excited polar semiconductors such as GaN. According to the calculations, there should not exist hot phonon bottleneck,
4. Authors claimed Auger heating is the mechanism at high carrier density. Basically, Auger heating will certainly affect the carrier population and thus possibly hot PL. Moreover, Auger heating will mostly occur at conduction band edge, then carriers are redistributed through further phonon scattering. There is not evidence for these.
5. Authors presented that free carrier Coulomb screening effect on HC cooling is negligible for MAPbI₃. For another possible origin of slow HC cooling – acoustic phonon upconversion, can authors comment it?
6. For the low temperature analysis, even ignore the temperature dependence of phonon scattering, the excitation energy is fixed, so the energy difference and initial temperature difference are the same. The reduction of temperature is essentially the same for different temperature.

Reviewer #3 (Remarks to the Author):

The authors present transient absorption and ab initio calculation results on MAPbI₃. From this, they extracted carrier temperature as a function of time in ps and setup a model that explains the dynamics of the carrier cooling, which they claim to be due to the interplay of hot phonon and Auger heating effects.

Although they were able to fit their experimental results (including excitation and temperature dependence) to the proposed model, one critical contribution was neglected for convenience as admitted by the authors, that is, the role of hot holes. These carriers are very much present in the system and are unambiguously identified in many published works. This cannot be easily ignored since the effective mass and mobility are very similar in these systems. Therefore, the dynamics of one charge specie unavoidably affect the other.

Moreover, the finding that Auger heating effects is present at higher excitation densities is not a unique claim as many systems, e.g. organic photovoltaic and quantum dot systems, have been reported to have the same behavior. In general, at elevated carrier densities, Auger process becomes dominant mechanism of recombination.

This reviewer also finds a disconnect between ab initio calculation results and experiments. Phonon modes obtained has already been known in several previous works and the attempt to relate it to the experimental results without additional evidence, maybe IR spectra, etc., makes the paper less convincing.

Due to the above considerations, this reviewer cannot recommend the publication of the work.

REVIEWERS' COMMENTS:

Reviewer #1 (Remarks to the Author):

I appreciate the author's sincere and deep consideration for my comments. There additional discussion in the manuscript makes the worth of their finding much clearer. Now I recommend this article to be published in nature communication.

Reviewer #2 (Remarks to the Author):

Authors have comprehensively considered and addressed the referees' comments. Accordingly, they significantly improved the manuscript. This revised manuscript present more detailed experimental and theoretical evidence to support the proposed mechanism for slow hot carrier cooling in perovskite, which is very useful for the understanding and development of the high efficiency hot carrier solar cells. I recommend publishing this manuscript in Nature Communications.

Reviewer #3 (Remarks to the Author):

The authors addressed all of my concerns. A better reformulation of the role of holes in hot carrier cooling would help readers better understand their arguments.

Reviewers' comments are highlighted in brown

Our responses are in black

The additional or revised sentences in manuscript are highlighted in blue

Reviewer #1 (Remarks to the Author):

This manuscript reports the hot-carrier cooling mechanism of MAPbI₃ by analysis of transient absorption spectroscopy in two different ranges of carrier concentration. At moderate hot-carrier concentration, the hot carrier temperature is well fitted with the model of electron-phonon interaction including hot phonon bottleneck effect. At higher concentration, Auger heating is also prominent and the lifetime of hot carrier becomes longer. This paper is very useful for the development of photovoltaics with higher performance. However, I'd like to ask the authors some further discussion before accepted.

Response: We are delighted that reviewer 1 shares our enthusiasm of the discoveries. We are greatly appreciative of the reviewer's critical comments, which are very helpful for improving our manuscript. We have carefully considered the referee's comments and below are our replies:

Comment 1) I have one question about LO phonon lifetime of MAPbI₃. This manuscript reported the lifetime is about 0.6 ps at room temperature. It is argued that this value is longer than the typical value of "electron-phonon" lifetime ~13 fs and hence it is the origin of hot phonon bottleneck. For example, as mentioned in the manuscript, InP and GaAs have longer lifetimes than 2 ps.

Ref 30 in this article (PCCP 18 27051(2016)) describes that "the short phonon lifetimes in MAPbX₃ imply a high degree of anharmonic phonon-phonon coupling". This statement seems to be opposed to the author's statement that "The efficient Klemens channel for LO phonon decay is thus hindered" in line 325. In conclusion, I'd like to ask you some more detailed discussion about the mechanism of hot phonon bottleneck.

Response 1: We appreciate the reviewer's comments and insightful questions about the potential inconsistencies/ambiguity of our results with the findings reported in the literature. The presence of a hot phonon bottleneck is not simply determined by the absolute phonon lifetime τ_{ph} , but by the competition between the phonon decay rate ($k_{ph} = 1/\tau_{ph}$) and the electron-phonon scattering rate (k_{e-p} which is proportional to the energy loss rate), that are in turn dependent on the carrier density (*Sov. Phys. JETP* **67**, 193 (1988)). In general, for polar semiconductors, which has efficient Klemens channel for LO phonon relaxation (*e.g.*, for GaAs *Phys. Rev. B* **43**, 7231 (1991)), the hot phonon bottleneck effect becomes obvious when there is a large non-equilibrium LO phonon population following intense excitation with carrier concentration well above 10^{18} cm⁻³. For other semiconductors, which the Klemens channel is blocked for LO phonon relaxation (*e.g.*, InP, *Phys. Rev. B* **47**, 13233, (1993)), the hot phonon effect can significantly slow down the hot carrier cooling process even under moderate excitation conditions (*i.e.*, analogous to MAPbI₃) because the LO phonon relaxation time is sufficiently long for a substantial build-up of hot phonons that reduces the net LO phonon emission. Thus, the hot phonon bottleneck effect can occur both in the case for highly excited semiconductors with efficient Klemens channel for LO phonon relaxation (1st example); as well as for the case of moderately excited semiconductors with suppression of the Klemens channel for LO phonon relaxation (2nd example).

We acknowledge that the short phonon lifetime indicates a high degree of anharmonic phonon-phonon coupling. However, it is important that the term "...the short phonon lifetimes in MAPbX₃..." must be

taken in context. The relatively shorter LO phonon lifetime of MAPbI₃ compared to other semiconductors is attributed to the unlocking rotation of MA ions, causing homogeneous broadening (ref 30 - (*PCCP* **18** 27051(2016))). Furthermore, it should also be noted that the estimate of LO phonon lifetime performed by the authors in ref 30 is strongly dependent on the experimental data as well as the fitting procedure (*i.e.*, deconvolution of Lorentzian contribution). The phonon lifetime estimated by the authors in article is based on an empirical formula: $\tau = \frac{\hbar}{FWHM}$. Hence, the estimate of the peak FWHM by the authors is influenced by the laser pulse and excitation fluence used. Meanwhile, since the LO phonon energy is only approximately 13 meV (< 26 meV for room temperature thermal energy) (*Nat. Commun.* **7**, 11755 (2016)), the experimental measurements of the peak FWHM (from Raman or IR spectra of MAPbI₃ at room temperature) would have a larger degree of uncertainty. An upper limit estimate of 26 meV divided by 13 meV, would mean that the authors' reported phonon lifetime of 0.1 ps could be smaller by as much as a factor of 2 (*i.e.*, 0.2 ps instead of 0.1 ps). Nonetheless, the estimated phonon lifetime of ~0.1 ps (or 100 fs) is still an order of magnitude larger than the electron-LO phonon scattering time constant (~ 13 fs), indicating the presence of hot phonon effect. In view of all the above considerations, our fitted LO phonon lifetime of ~0.6 ps is in good correspondence with the estimated phonon lifetime of ~0.1 ps reported by the authors in ref 30.

To make the manuscript clearer, we have added the following statements on page 13, paragraph 3, line 3 of the revised manuscript:

Regarding the hot phonon effect (a), there is a complex interplay of two factors: (1) the carrier density and (2) the LO phonon lifetime. The presence of a hot phonon bottleneck is not simply determined by the absolute phonon lifetime τ_{ph} , but by the competition between the phonon decay rate ($k_{\text{ph}} = 1/\tau_{\text{ph}}$) and the electron-phonon scattering rate ($k_{\text{e-p}}$ which is proportional to the energy loss rate), that are in turn dependent on the carrier density. In general, under low excitation conditions ($n < 10^{17} \text{ cm}^{-3}$), the hot phonon effect is negligible and the energy loss rate of the HC is independent of carrier density. However, under highly excited conditions with carrier density well above 10^{18} cm^{-3} , significant slow HC cooling can be observed due to the presence of a large non-equilibrium LO phonon population, which leads to reduced net LO phonon emission. This highly excited regime is characterized by an obvious reduced energy loss rate with carrier density, *i.e.*, slower HC cooling. For the second factor, the long LO phonon lifetime can enhance hot phonon effect by lengthening the LO phonon decay given that HC cooling is a cascade process between electron-LO phonon and anharmonic phonon-phonon interactions. For lead halide perovskites, the hot phonon effect is non-trivial even at moderate carrier densities (Supplementary Table 1). Our model yields a fitted LO phonon lifetime of around 0.6 ± 0.1 ps at room temperature, which is much longer than typical electron-LO phonon scattering time constant of approximately 13 fs (See Supplementary Information for details).

Comment 2) As described in line 330-334, the large LO phonon population coming from the small LO phonon energy is plausible. In addition, the authors may be able to mention the electronic structure of MAPbI₃. For example, some papers argue that the low density of states of valence and conduction bands are related to the long electron-phonon lifetime and the high open-circuit voltage. For example, please check two papers below. - *Nano Lett*, **15**, 3103 (2015)

- *J. Phys. Chem. C*, **121**, 1455 (2017)

The density state must be related to the effective mass in your model. How is the impact of the light mass on the hot phonon bottleneck?

Response 2: We thank the reviewer for the constructive comments and pointing us to the relevant publications relating to the electronic structure of MAPbI₃, electron-phonon lifetime and the high open-circuit voltage (*Nano Lett*, **15**, 3103 (2015); *J. Phys. Chem. C*, **121**, 1455 (2017)). Accordingly, the electronic structure of MAPbI₃ would have little effect on the V_{oc} and hot carrier cooling. Meanwhile, the density of states (DOS) plays a significant role in the semiconductor properties of MAPbI₃. Low DOS of valence and conduction band can lead to high V_{oc} given that the relation between V_{oc} and the effective density of states: $V_{oc} = \frac{1}{e} \left[E_g + k_B T \ln(np) - k_B T \ln(N_c N_v) \right]$, where E_g is the bandgap, n and p are the electron and hole density, N_c and N_v are the effective density of the conduction and valence band. It is well known that low DOS of valence and conduction bands will lead to small effective mass and high charge carrier mobility. On the other hand, low density states of the conduction and valence band will result in fewer available relaxation pathways for the hot carriers and retardation of hot carrier cooling (*Nano Lett*, **15**, 3103 (2015)). From a physical point of view, the effective mass of the electron can affect the hot carrier cooling via modifying the energy loss rate of the hot carriers. Since the energy loss rate of the hot carrier can be expressed by:

$$P_0 = eE_0 \left(\frac{2\hbar\omega_{LO}}{m} \right)^{\frac{1}{2}} = \frac{m^{1/2} e^2}{\pi\epsilon_0 \hbar^2} \left(\frac{\hbar\omega_{LO}}{2} \right)^{3/2} \left[\frac{1}{\epsilon_{Opt}} - \frac{1}{\epsilon_{Stat}} \right]$$

which means that the lighter the effective mass, the smaller the energy loss rate for the hot carriers. Thus, the lighter mass will lead to less efficient hot carrier relaxation and more significant hot phonon bottleneck effect.

To strengthen our manuscript, we added the following sentences in the manuscript on page 14, paragraph 1, line 27 of the revised main manuscript:

Furthermore, the low density of states (DOS) of the electronic structure in MAPbI₃ could also enhance the hot phonon effect (*Nano Lett*, **15**, 3103 (2015)). This is because a small DOS will lead to a reduction of the available relaxation pathways as well as small effective mass, which can aggravate the hot phonon effect by reducing the energy loss rate of the HC (Supplementary Fig. S13). Hence, the suppression of efficient Klemens channel for LO phonon decay, together with the large non-equilibrium LO phonon population and small DOS of the electronic structure, gives rise to the strong hot phonon bottleneck effect in lead halide perovskites.

and on page 26 and page 14 in the Supplementary materials:

Supplementary Note 6. Dependence on the electron effective mass

As for the dependence of hot carrier cooling on the effective electron mass, the scenario can be understood from the energy loss of the hot carriers in the absence of hot phonon effect. The energy loss rate of the hot carriers can be expressed by:

$$P_0 = eE_0 \left(\frac{2\hbar\omega_{\text{LO}}}{m} \right)^{\frac{1}{2}} = \frac{m^{1/2}e^2}{\pi\epsilon_0\hbar^2} \left(\frac{\hbar\omega_{\text{LO}}}{2} \right)^{3/2} \left[\frac{1}{\epsilon_{\text{Opt}}} - \frac{1}{\epsilon_{\text{Stat}}} \right] \quad (\text{S9})$$

which means that the lighter the mass, the smaller the energy loss rate for the hot carriers. Thus, a lighter mass will lead to less efficient hot carrier relaxation and more significant hot phonon bottleneck effect. To demonstrate the influence of the effective mass on hot carrier cooling, we calculate the hot carrier cooling dynamics and the relative non-equilibrium LO phonon population for $m_e = 0.15 m_0$ (*J. Mater. Chem. A*, **3**, 9208 (2015)), $m_e = 0.19 m_0$ (this work) and $m_e = 0.23 m_0$ (*J. Phys. Chem. Lett.* **4**, 4213 (2013)) at a carrier concentration of $n_0 = 4.2 \times 10^{18} \text{ cm}^{-3}$. As shown in *Fig. S13*, a lighter mass will result in a slower hot carrier cooling process. This is consistent with the low DOS of the valence band (*Nano Lett.* **15**, 3103 (2015)) in lead iodide perovskites APbI₃ (A can be CH₃NH₃, NH₂CH=NH₂ and Cs), which also leads to slow hot hole cooling.

Fig. S13 Influence of effective electron mass m_e on hot carrier cooling and non-equilibrium LO phonon population dynamics. A lighter mass will result in a slower hot carrier cooling process. The calculations were performed for different effective electron masses (a) $m_e = 0.15 m_0$, (b) $m_e = 0.19 m_0$, and (c) $m_e = 0.23 m_0$ at the same carrier concentration of $4.2 \times 10^{18} \text{ cm}^{-3}$.

Comment 3) I believe that the manuscript must be much more useful if the authors suggest parameters to differentiate MAPbI₃ from other semiconductors.

Response 3: We thank the reviewer for the constructive suggestion. To differentiate MAPbI₃ from other semiconductors in terms of hot carrier cooling, we have thus added the well-studied hot carrier cooling parameters of typical semiconductors in the Supplementary Table 1 and revised the statement on page 14, paragraph 1, line 5 of the revised main manuscript as shown below:

For lead halide perovskites, the hot phonon effect is non-trivial even at moderate carrier densities (Supplementary Table 1). Our model yields a fitted LO phonon lifetime of around 0.6 ± 0.1 ps at room temperature, which is much longer than typical electron-LO phonon scattering time constant of approximately 13 fs (See Supplementary Information for details).

Supplementary Table 1. List of parameters relating to the hot-phonon effect for different semiconductors. The studies were performed at room temperature unless stated otherwise in parenthesis in column 1.

Semiconductor	LO Phonon lifetime (ps)	Hot carrier cooling lifetime (ps)	Carrier density (cm ⁻³ or cm ⁻² for 3D/2D systems) *	LO phonon energy (meV)	Methods	Reference
GaAs	~ 2.1	~ 1.85	~ 2×10^{18}	~ 36	Transient transmission	Phys. Rev. B 54 , 14487, 1996
GaAs quantum wells	—	~ 3	~ 1×10^{12}	~ 36	fluorescence up-conversion	Phys. Rev. B 45 , 1450, 1992
InP	7.6	~ 1.5	~ 5×10^{17}	~ 43	Transient absorption & Time-resolved luminescence	Phys. Rev. B 47 , 13233, 1993
ZnO	~ 1.75	0.4 ~ 1.0	> 8×10^{18}	~ 70	Transient transmission spectroscopy	Appl. Phys. Lett. 87 , 023106, 2005
GaN	0.5 ~ 2.5	~ 0.6	~ 10^{18}	92	Time-resolved Raman	Appl. Phys. Lett. 74 , 711, 1999
Al _x Ga _{1-x} N/GaN	0.35	—	~ 5×10^{12}	~ 92	short-time-domain gated radiometric microwave noise	Phys. Rev. B 68 , 035338, 2003

					technique	
CdSe (8 K)	~ 5	~ 10	$\sim 2 \times 10^{17}$	~ 26.5	Fluorescence up-conversion (~ 2.5 ps)	Phys. Rev. B 51 , 14233, 1995
CdS	~ 0.6	~ 2	$\sim 3 \times 10^{17}$	~ 35	Fluorescence up-conversion	Phys. Rev. B 52 , 4728, 1995
MAPbBr ₃ nanocrystals	—	0.5~1	Less than $\langle N_0 \rangle \sim 0.1$ or $n_{0\text{avg}} \sim \sim 2.6 \times 10^{17}$	~ 15.3 (Nat. Commun. 7 , 11755, 2016)	Transient transmission spectroscopy	Nat. Commun. 8 , 14350, 2017
MAPbI ₃ thin films	~ 0.6	0.3~1.0	$\sim 6 \times 10^{17}$	~ 13	Transient transmission spectroscopy	This work

* Carrier density when obvious hot phonon effect is observed.

(As discussed in our earlier work (*Nat. Commun.* **8**, 14350, 2017), the measured hot-phonon lifetime could also be limited by the time-resolution of the experimental techniques used, thereby yielding artificially longer lifetimes that are limited by the system temporal response rather than its intrinsic hot-phonon lifetime. Hence, due care must be taken for a fair comparison of the reported values in the literature.)

Reviewer #2 (Remarks to the Author):

Recently, slowed cooling of hot carrier has been extensively observed, and some mechanisms for hot phonon bottleneck have been proposed, including screening effect and acoustical-optical phonon upconversion. Given the significantly potential applications of hot carriers, such as high efficiency hot carrier solar cells, and photo-catalysis, a clear mechanism of the hot carrier in perovskite bulk and nanostructures are crucially important. In this study, authors experimentally observed fluence-dependent hot-carrier dynamics in perovskite film by transient absorption and time-resolved PL. They correlated the hot carrier effect with the theoretical modelling and first principles calculations. They proposed at low carrier density the hot carrier cooling is slowed down dominantly by the hot phonon bottleneck due to suppressed Klemens relaxation pathway (large energy separation between the LO and LA phonon branches), so observed cooling time of $\sim 0.6 \pm 0.1$ ps. At a high carrier density, Auger heating reduces the cooling rate; up to 50 ps cooling time was observed. Basically, the manuscript is significant, well organized and clearly presented. Authors have presented significant experimental and theoretical evidences to support their conclusions. However, I do not think really convinced, as some examples in the following. I cannot recommend accepting for publication until a significantly revision and improvement.

Response: We thank the reviewer for his critical review of our manuscript and we greatly appreciate his feedback to help us strengthen our manuscript.

Comment 1:

In terms of the slowed cooling of ~ 600 fs, it is roughly in the range of a normal cooling time without any hot phonon bottleneck for most semiconductors (experimental observations). Authors compared and claimed delayed slowed cooling time based on calculated 15 fs of cooling time, for the case without hot phonon effect. It is not valid. This cooling time of 15 fs is also inconsistent with most observations of perovskites.

Response: We thank the reviewer for his comments. However, we respectfully disagree with the reviewer about his first comment that the “slowed cooling of ~ 600 fs is roughly in the range of a normal cooling time *without* any hot phonon bottleneck for most semiconductors (experimental observations)”. Here the 0.6 ps or 600 fs refers to the LO phonon lifetime and not the cooling lifetime. This was stated on page 8, paragraph 2, line 5 of the main manuscript: “*The LO phonon lifetime τ_{ph} , which is associated with the anharmonic interactions between LO phonon and acoustic phonons, is in general temperature dependent and varies for different phonon-phonon coupling interactions*”. Perhaps there has been some misunderstanding.

We also respectfully disagree with his comment that the “... cooling time of 15 fs is also inconsistent with most observations of perovskites”. We suppose that the reviewer is referring to the calculated 13 fs lifetime (since there is no 15 fs lifetime mentioned anywhere in our manuscript). We would like to reiterate that the 13 fs was calculated based on a hypothetical situation that the hot phonon bottleneck mechanism is absent. In a recent work, Yang, *et al.*, (*Nat. Photon.* **10**, 53, (2016)), the authors had also estimated a cooling lifetime of ($\sim 30 \pm 3$ fs) in the absence of a hot phonon bottleneck. The timescale is consistent with ours. Hence, we disagree that there is any inconsistency or contradiction.

A related question on the 0.6 ps LO phonon lifetime and the 13 fs “electron-phonon lifetime” was raised by Reviewer 1 (Comment 1). Please refer to our response (Response 1) to that question.

In view of the reviewer's concerns and the potential ambiguity, we have also made the following revisions on page 2, paragraph 1, line 10 in the abstract to make our manuscript clearer:

“At moderate carrier concentrations ($\sim 10^{17}$ - 10^{18} cm^{-3}), carrier cooling is mediated by polar Fröhlich electron-phonon interactions through zone-center delayed LO phonon emissions (*i.e.*, with phonon lifetime $\tau_{\text{LO}} \sim 0.6 \pm 0.1$ ps) induced by the hot phonon bottleneck.”

Comment 2:

2. MAPbI₃ has been proposed a second valence band, this will affect the initial relaxation and thus the hot carrier relaxation, author should also consider the possible influence;

Response 2: We thank the reviewer for his comment. Indeed, this has been taken into consideration. In our first work which first reported this phenomenon of slow hot carrier cooling (*Science* **342**, 344 (2013)), we had excited MAPbI₃ thin films to above this second valence band (with 400 nm excitation). The presence of this second valence band (at ~ 500 nm) yielded a slow hot carrier cooling lifetime of $\sim 0.4 \pm 0.1$ ps. In this work, we resonantly excite this this second valence band (with 2.48 eV or ~ 500 nm pulses). The carrier cooling lifetime is consistent with our earlier reported lifetimes.

Comment 3:

This is well-known that hot phonon effect that is commonly observed in highly excited polar semiconductors such as GaN. According to the calculations, there should not exist hot phonon bottleneck.

Response 3: We thank the reviewer for his comment. Indeed the reviewer is correct about the “... hot phonon effect that is commonly observed in highly excited polar semiconductors such as GaN.” Here, MAPbI₃ perovskite is polar. Our calculations are based on this fact which we had stated on (*line 126* of the original manuscript - “*To simplify the carrier-phonon interaction process, we consider a direct bandgap polar semiconductor with parabolic band structure following femtosecond pulse excitation*”).

Comment 4:

Authors claimed Auger heating is the mechanism at high carrier density. Basically, Auger heating will certainly affect the carrier population and thus possibly hot PL. Moreover, Auger heating will mostly occur at conduction band edge, then carriers are redistributed through further phonon scattering. There is not evidence for these.

Response 4: We thank the reviewer for his comment. We would like to point out that we had indeed observed a significant decay of the bandedge photobleaching kinetics (please refer to Supplementary Fig. S5). This is clear evidence that strong carrier-carrier interactions (Auger heating) are present and need to be taken into account for the hot carrier cooling process.

To make it clearer, we have revised Fig. S5 in Supplementary materials on page 6.

Fig. S5 (a) Transient absorption spectra of perovskite films with different probe delay times. The spectrum was collected at pump energy of 2.48 eV with carrier density $n_0 = 10.4 \times 10^{18} \text{ cm}^{-3}$. Green dashed line serves as vertical guide line for the eye. Red line: fast red-shift of TA valley. Blue: slow red-shift of TA valley. Inset shows the red-shift of the center of the TA valley with delay time (b) Normalized band-edge photobleaching kinetics, square dots: experiment data, red line: fitting with third-order exponential decay. The fitted Auger lifetime is $\sim 137 \text{ ps}$ which is quantitatively consistent with the estimated value from $\tau_{\text{Auger}} = 1/k_3 n^2 \approx 119 \text{ ps}$.

To further prove the effect of Auger heating on hot carrier cooling, we have now included the time-resolved photoluminescence (TRPL) at a high carrier concentration $\sim 3 \times 10^{19} \text{ cm}^{-3}$, as shown in the figure below. Therefore, we add the following new statements to the Supplementary Fig. S12 on page 13, and modified the main manuscript on page 12, paragraph 1, line 15 to further strengthen our manuscript:

This deduction is supported by the observations of rapid shortening of the band-edge PB kinetics that is typical for multi-particle Auger recombination process (Supplementary Fig. S5) and the persistent higher energy tail of hot PL (Supplementary Fig. S12).

Fig. S12 (a) TRPL dynamics of MAPbI₃ films excited at 2.48 eV with carrier concentration of $\sim 3 \times 10^{19} \text{ cm}^{-3}$, the solid line is a single exponential decay with fitted lifetime of $\sim 115 \text{ ps}$. (b) Evolution of the hot photoluminescence spectra at representative delay times longer than 10 ps when Auger heating occurs (streak camera data). The narrowing of the high energy tail with longer delay times indicates that the hot carrier cooling is further slowed down by the Auger heating effect. The solid lines were fitted using: $I(E) \propto [g(E)f(E)]^2$, where

$g(E) = \frac{1}{2\pi^2} \frac{(2m^*)^{3/2}}{\hbar^3} \sqrt{E - E_g}$ is the joint density of states for the symmetric and parabolic bands, $f(E)$ is the Fermi-Dirac distribution function. Together with the observations of rapid shortening of the band-edge PB kinetics (Supplementary Fig. S5), multi-particle Auger recombination process is present.

Comment 5:

Authors presented that free carrier Coulomb screening effect on HC cooling is negligible for MAPbI₃. For another possible origin of slow HC cooling – acoustic phonon up-conversion, can authors comment it?

Response 5: We thank the reviewer for his question. The authors in *Nat. Commun.*, **8**, 14120 (2017) discovered a comparatively stronger phonon bottleneck effect in hybrid organic-inorganic perovskites than in their fully inorganic caesium counterparts. In contrast to the caesium-based system, a 10 times slower carrier-phonon relaxation rate was observed for the hybrid perovskites, which they had attributed to the larger acoustic-optic phonon up-conversion efficiency present in the latter.

This acoustic-optical phonon up-conversion mechanism proposed by the authors was primarily based on the findings of another work (*Phys. Rev. B* **91**, 054111 (2015)) – that is for a (thermally) insulating layered (two-dimensional (2D)) hybrid perovskite system ($(\text{C}_6\text{H}_5\text{CH}_2\text{CH}_2\text{NH}_3)_2\text{CuCl}_4$), where its thermal coupling constant was found to be $g \sim 0.4 \times 10^{17} \text{ W/Km}^3$). In contrast, MAPbI₃ is a three-dimensional (3D) hybrid perovskite system. The lattice thermal conductivity of MAPbI₃ is 0.5 W/Km (*J. Phys. Chem. Lett.*, 2014, 5 (14), pp 2488–2492). Using a lattice constant of 8.8 \AA (*J. Phys. Chem. A* **119**, 11033 (2015)) for MAPbI₃, the thermal coupling constant can be estimated to be $g \sim 6.5 \times 10^{17} \text{ W/Km}^3$. This estimated value is approximately 1 order larger than the 2D system of the work that is cited by the authors. Furthermore, in another recent publication (*Nat. Photon.* **10**, 115–121 (2016)) on laser cooling of 2D and

3D halide perovskites, the measured maximum cooling for 2D is about 300 K m W^{-1} while that for 3D halide perovskites is about 35 K m W^{-1} . These latter results are in agreement with our estimate presented above (*i.e.*, 1 order difference between 3D and 2D halide perovskites). Hence, the 3D MAPbI₃ system does not possess the same thermally insulating qualities as the 2D layered (C₆H₅CH₂CH₂NH₃)₂CuCl₄ system. The underlying assumption in the acoustic-optical phonon up-conversion mechanism proposed in *Nat. Commun.*, **8**, 14120 (2017) was based on 3D and 2D halide perovskites possessing similar thermal properties. This need not be the case.

However, we hold a different view of the hot carrier cooling mechanism in MAPbI₃ perovskites. The first stage of hot carrier cooling (within $\sim 300 \text{ fs}$) is characterized by hot carrier thermalization (which reaches a quasi-equilibrium temperature T_e within $\sim 100 \text{ fs}$), followed by fast LO phonon emission and build-up of LO phonon population. The second stage ($0.3\sim 2 \text{ ps}$) of hot carrier cooling is however slowed down by the presence of large hot LO phonon population. This stage becomes obvious when the perovskites were moderately excited ($\sim 10^{18} \text{ cm}^{-3}$), which is characterized by the carrier density dependent energy loss rate. The third stage of long-lived ($\sim 100 \text{ ps}$) slow hot carrier cooling is dominated by the Auger heating process. This stage becomes significant especially when the semiconductors are highly excited ($\sim 10^{19} \text{ cm}^{-3}$). It is well known that the Auger recombination coefficient ($\sim 5 \times 10^{-41} \text{ cm}^6 \text{ ps}^{-1}$) in MAPbI₃ is very large, and is inversely dependent on the band gap. Hence, the Auger heating effect cannot be ignored when the perovskites are highly excited ($\sim 10^{19} \text{ cm}^{-3}$). In short, at moderate excitations, slow hot carrier cooling in perovskites is due to the hot phonon effect; whereas at highly excitation conditions, slow hot carrier cooling is dominated by the interplay between hot phonon and Auger heating effect.

Furthermore, our findings show evidence for the suppression of the Klemens channel. The Klemens channel for LO phonon relaxation occurs when both momentum and energy are conserved: $\mathbf{k}_1 + \mathbf{k}_2 = \mathbf{k}_0$ and $\hbar\omega_o(\mathbf{k}_0) = \hbar\omega_A(\mathbf{k}_1) + \hbar\omega_A(\mathbf{k}_2)$. The Klemens channel will be blocked when the LO phonon energy is larger than twice of the LA phonon energy, *i.e.* LO > 2LA (*Physica E*, **42**, 2862 (2010) pp 2863). Our phonon calculations present clear LO phonon and LA phonon bands in MAPbI₃ (main manuscript Figure 4). The energy separation between them is more than 2 times larger (*i.e.*, LO phonon energy ($\sim 13 \text{ meV}$) and LA phonon energy ($\sim 2.5 \text{ meV}$)). These findings support the blocking of the Klemens channel.

In summary, we hold a different view of the mechanism of the slow hot carrier cooling. Our findings support the suppression of the Klemens channel and the Auger heating process as the origins to the slow hot carrier cooling effect in perovskites.

In view of the reviewer's concern, we have added a comment in the main manuscript on page 16, paragraph 2, line 8:

Our findings support the suppression of the Klemens channel and the Auger heating process as the origins to the slow hot carrier cooling effect in perovskites. This contrasts with a recent report (*Nat. Commun.*, **8**, 14120 (2017)) proposing the acoustic-optical phonon up-conversion mechanism (*Phys. Rev. B* **91**, 054111 (2015)) being responsible for the phonon bottleneck effect. Their underlying assumption was based on three-dimensional and two-dimensional halide perovskites possessing similar thermal properties, which need not be the case (*Nat. Photon.* **10**, 115–121 (2016)).

Comment 6:

For the low temperature analysis, even ignore the temperature dependence of phonon scattering, the excitation energy is fixed, so the energy difference and initial temperature difference are the same. The reduction of temperature is essentially the same for different temperature.

Response 6: We thank the reviewer for the comments. However, we respectfully disagree with them. At different temperatures, the equilibrium state of the hot carriers is different. It should be noted that the hot carrier cooling is a cascade process between electron-LO and phonon-phonon interactions: the hot carriers lose their excess energy by net LO phonon emission, and the LO phonons thermalize by anharmonic phonon-phonon interactions. If the LO phonon lifetimes becomes longer, enhanced hot phonon effect will be observed since the LO phonon takes more time to decay which can lead to reduced net LO phonon emission and slower hot carrier cooling. The increase of LO phonon lifetime with reduced temperature can be derived by from the following relations:

The population rate of the LO phonon (*Phys. Rev.* **148**, 845 (1966) & *J. Phys.: Condens. Matter* **8**, L511 (1996)), irrespective of the Klemens channel or Ridley channel can be derived:

$$dn(\omega, T)/dt = -n(\omega, T)[1 + n_1(\omega_1, T) + n_2(\omega_2, T)]/\tau_0$$

where τ_0 is the lifetime of LO phonon at zero temperature, $n(\omega, T) = [\exp(\hbar\omega/k_B T) - 1]^{-1}$ is the phonon occupation number. The effective lifetime of LO phonon is thus:

$$\tau = \left| \frac{dn}{n} \right|^{-1} = \frac{\tau_0}{1 + n_1(\omega_1, T) + n_2(\omega_2, T)} \propto \frac{1}{T}$$

Therefore, the LO phonon lifetime increases with decreasing temperature. The slower hot carrier cooling with decreasing temperature is therefore consistent with the increase of LO phonon lifetime with the reduced temperature (*Phys. Rev. B* **28**, 7040, (1983), *Phys. Rev. B* **57**, 15337, (1998)) stems from the decrease of phonon-phonon scattering rates due to the associated decrease in phonon occupancy (*Phys. Rev. B* **83**, 205411 (2011)). Furthermore, our experimental results (Figure 2b of main manuscript) clearly show a dependence of the cooling as a function of temperature which we had reproduced below (see figure R1):

Figure R1 HC cooling governed by the hot phonon bottleneck effect at carrier densities around 10^{18} cm^{-3} . (a) HC cooling of MAPbI₃ films excited with same $n_0 = 4.2 \times 10^{18} \text{ cm}^{-3}$ but varying excitation energy. (b) Temperature dependent HC cooling dynamics of MAPbI₃ films excited at 2.48 eV with initial carrier density $n_0 = 5.5 \times 10^{18} \text{ cm}^{-3}$. The dashed lines are calculated HC cooling dynamics using our model (see main text).

Reviewer #3 (Remarks to the Author):

Comment: The authors present transient absorption and ab initio calculation results on MAPbI₃. From this, they extracted carrier temperature as a function of time in ps and setup a model that explains the dynamics of the carrier cooling, which they claim to be due to the interplay of hot phonon and Auger heating effects. Although they were able to fit their experimental results (including excitation and temperature dependence) to the proposed model, one critical contribution was neglected for convenience as admitted by the authors, that is, the role of hot holes. These carriers are very much present in the system and are unambiguously identified in many published works. This cannot be easily ignored since the effective mass and mobility are very similar in these systems. Therefore, the dynamics of one charge specie unavoidably affect the other.

Response: We thank the reviewer for the critical comments. However, we respectfully disagree with the reviewer's comment that the role of the holes is critically neglected for convenience in our work. We had only assumed that the extracted HC temperature to be the electron's ($T_c \approx T_e$) for convenience. We had not neglected the role of the hot holes. Perhaps there had been some misunderstanding.

On the contrary, the influence of both species (electrons and holes) are well taken into consideration in our analysis of the TA spectroscopy data. TA signals originate from a convolution of both the electrons and holes. This was clearly stated in our original manuscript at *line* 106, which is reproduced below.

“Given that the electrons and holes contribute to the TA response, the electrons' excess energies are almost the same as that of the holes given their similar effective masses ($m_e = 0.19 m_0, m_h = 0.25 m_0$). Therefore, for convenience, we assume that the extracted HC temperature to be the electron's ($T_c \approx T_e$), which is reasonable.”

Being fermions, electron and holes are described by the Fermi-Dirac distribution. We assumed that the extracted HC temperature to be the electron's ($T_c \approx T_e$) for convenience, given their similar effective masses. Our approach of treating the extracted HC temperature to be the electron's is a classic treatment of electron-phonon scattering (*The Physics of Two Dimensional Electron Gas—Electron-phonon interactions in 3D, pp185 (1986)* by J.T. Devreese and F.M. Peeters).

Comment: Moreover, the finding that Auger heating effects is present at higher excitation densities is not a unique claim as many systems, e.g. organic photovoltaic and quantum dot systems, have been reported to have the same behavior. In general, at elevated carrier densities, Auger process becomes dominant mechanism of recombination.

Response: We thank the reviewer for the comment. However, we again respectfully disagree with this sweeping statement about Auger recombination in many systems. It should be noted here that our work is the first report proposing an elegant mathematically tractable hot phonon model with Auger heating effects that satisfactorily accounts for the slow hot carrier cooling phenomena in MAPbI₃ films over a broad range of carrier densities, without needing to invoke intriguing concepts (e.g., polaron screening, acoustic-optical phonon upconversion). Prior to our work, there is no report on the Auger heating effect on the slow hot carrier cooling in bulk halide perovskite systems. Moreover, previous works (*Nat. Photon.* **10**, 53 (2016) & *Nat. Commun.* **8**, 14120 (2017)) do not address the fluence dependent hot carrier cooling process. The mechanisms of slow hot carrier in halide perovskites are still highly controversial in the perovskite field. Our work therefore provides crucial insights that would help clear this puzzle. A clear understanding of the origins and mechanisms of the slow hot carrier cooling phenomenon in perovskites have significant relevance not only for basic curiosity but also for progressive development of perovskite optoelectronic applications.

Comment: This reviewer also finds a disconnect between ab initio calculation results and experiments. Phonon modes obtained has already been known in several previous works and the attempt to relate it to the experimental results without additional evidence, maybe IR spectra, etc., makes the paper less convincing.

Response: We thank the reviewer for his question. Indeed, the reviewer is correct that phonon modes obtained has already been known in several previous works. However, we would like to point out that experimental measurements and DFT calculations of the IR spectra for MAPbI₃ have also been reported by several groups (*J. Phys. Chem. C* **119**, 25703 (2015), *Phys. Rev. B* **92**, 144308 (2015), *Mater. Horiz.* **3**, 613 (2016), *ChemSusChem*, **9**, 2994 (2016)). Our calculated LO modes (around 12 meV) are in excellent agreement with the reported experimental values (~ 13 meV). Hence, we again respectfully disagree with the reviewer's view that there is a disconnect between ab initio calculation results and experiments

In view of the reviewer's concerns, we have now performed additional DFT calculations and Raman measurements for the MAPbI₃ films and single crystals. So far, several groups have reported the Raman spectra of MAPbI₃, e.g., *J. Phys. Chem. Lett.* **5**, 279 (2014), *J. Phys. Chem. Lett.* **6**, 401 (2015), *Phys. Chem. Chem. Phys.* **18**, 27051 (2016), with the respective LO phonon energy around 104.5 cm⁻¹ (~ 13 meV), 130 cm⁻¹ (~ 16 meV) and 97 cm⁻¹ (~ 12 meV), which are consistent with our measured Raman spectra (LO ~ 12.5~14 meV) and DFT calculations (See figure below). The consistency between our additional theoretical and experimental work with the reported results has further strengthened our manuscript and we are now even more confident of our findings. Hence, to further strengthen our manuscript, we have added the following discussions in the main manuscript on page 14, paragraph 1, line 15 and supplementary materials on page 16 as shown below.

Our measurements and DFT calculations (Supplementary Fig. S15) are in good agreement with literature reports (*J. Phys. Chem. Lett.* **6**, 401 (2015), *Phys. Chem. Chem. Phys.* **18**, 27051 (2016), *Phys. Rev. B* **94**, 094301 (2016)).

Fig. S15 Comparison of experimental measurements with the DFT calculations. Experimental measurements of Raman spectra for tetragonal MAPbI₃ films (green line) and single crystal (blue line) with an excitation wavelength of 532 nm at 170 K. The LO phonon modes of Pb²⁺ and I⁻ vibration lies at around 71.6 cm⁻¹ (~ 9 meV), 101 cm⁻¹ (~12.5 meV) and 114 cm⁻¹ (~ 14 meV), in agreement with the recent reports (*J. Phys. Chem. Lett.* **6**, 401 (2015), *Phys. Chem. Chem. Phys.* **18**, 27051 (2016)). The other two peaks centered around 27 cm⁻¹ and 44 cm⁻¹ correspond to the TO phonon modes. The measured LO phonon position corresponds to the upper yellow zone in figure S17. The theoretical Raman spectra were calculated by convolving the phonon modes with Lorentzian function with a FWHM of 8 cm⁻¹.

Reviewers' comments are highlighted in brown

Our responses are in black

The additional or revised sentences in manuscript are highlighted in blue

Reviewer #1 (Remarks to the Author):

I appreciate the author's sincere and deep consideration for my comments. There additional discussion in the manuscript makes the worth of their finding much clearer. Now I recommend this article to be published in nature communication.

Response: We are delighted with the reviewer's comments and all the valuable feedback. We would like to thank him/her for his/her time in strengthening our manuscript for publication.

Reviewer #2 (Remarks to the Author):

Authors have comprehensively considered and addressed the referees' comments. Accordingly, they significantly improved the manuscript. This revised manuscript present more detailed experimental and theoretical evidence to support the proposed mechanism for slow hot carrier cooling in perovskite, which is very useful for the understanding and development of the high efficiency hot carrier solar cells. I recommend publishing this manuscript in Nature Communications.

Response: We are delighted with the reviewer's comments and all the valuable feedback. We would like to thank him/her for his/her time to help strengthen our manuscript for publication.

Reviewer #3 (Remarks to the Author):

Comment: The authors addressed all of my concerns. A better reformulation of the role of holes in hot carrier cooling would help readers better understand their arguments.

Response: We thank the reviewer for all his valuable feedback and constructive suggestions. We would like to reiterate that the transient absorption response is the summation of the contribution from both the electron and the hole since $\Delta A(E) = -A_0(E)[f_e(E_c) + f_h(E_v)]$. The contributions of the electron and the hole to the hot carrier cooling process will be roughly equal (*Nat. Photon.*, 10, 53-59(2016), *Physica E*, 14, 115-120(2002)), since the effective electron mass and effective hole mass are similar. Inclusion of this difference only slightly changes the prefactor of 2 for the exponential in $\Delta A(E) = -A_0(E)[f_e(E_c) + f_h(E_v)]$. Most importantly, this will not affect the derived hot carrier temperature (that is embedded within the exponential term). Therefore, the extracted hot carrier temperature can be assumed to be the electron's for convenience.

This similarity in effective masses also aptly highlights a distinct advantage of perovskites over conventional semiconductors for hot carrier applications. In typical semiconductors, their electron effective mass is much lower than that of the hole. This will result in faster hot hole cooling compared to hot electron cooling. In contrast, the more “balanced” hot electron and hot hole cooling in perovskites will be more amenable for the development of practical hot-carrier optoelectronic devices.

In view of the reviewer's concern, we have revised the following statements to highlight these points on page 6, paragraph 1, line 8 of the revised manuscript:

Both electrons and holes contribute to the TA response. Given their similar effective masses ($m_e = 0.19 m_0$, $m_h = 0.25 m_0$), their contributions to the hot carrier cooling process will be roughly the same (*Nat. Photon.*, 10, 53-59(2016), *Sci. Rep.* 4, 4467(2014), *Physica E*, 14, 115-120(2002)). Therefore, for convenience, we assume that the extracted HC temperature to be the electron's ($T_c \approx T_e$), which will not change the conclusions here (see Supplementary Note 4).

And we add the statements in page 25, paragraph 2, line 4 of the revised Supplementary Materials as follows:

where $A_0(E)$ is steady state linear absorbance. One therefore will always obtain a negative ΔA contributed by both the electrons and the holes. Meanwhile, from the equation we can see that the profile of ΔA correlates straightforwardly with Fermi distributions of the electrons and the holes. Given their similar effective masses (*Sci. Rep.* 4, 4467(2014)) ($m_e = 0.19 m_0$, $m_h = 0.25 m_0$), their contributions to the hot carrier cooling process will be roughly the same (*i.e.*, $f_e \approx f_h$). Inclusion of this difference only slightly changes the prefactor of 2 for the exponential in Eqn 5. Most importantly, this will not affect the derived hot carrier temperature (that is embedded within the exponential term). Hence, for convenience, we assume that the extracted HC temperature to be the electron's ($T_c \approx T_e$), which will not change the physics and the conclusions here.

Furthermore, we also added a few statements on page 13, paragraph 2, line 1 to the revised manuscript to better emphasize the significance of the similar electron and hole effective masses for the development of practical hot-carrier applications:

The similarity of perovskite's electron and hole effective masses aptly highlights its distinct advantage over conventional semiconductors for hot carrier applications. In typical semiconductors, their electron effective mass is much lower than that of the hole. This will result in faster hot hole cooling compared to hot electron cooling. In contrast, the more "balanced" slow hot electron and hot hole cooling in perovskites will be more amenable for developing practical hot-carrier optoelectronic devices. In semiconductors, possible origins of slow HC cooling include: hot phonon effect, Coulomb screening effect, and Auger heating effect.

And another paragraph on page 29, paragraph 1 of the revised Supplementary materials:

In light of the relation between the effective mass and the hot carrier cooling rate, the similarity of perovskite's electron and hole effective masses aptly highlights its distinct advantage over conventional semiconductors for hot carrier applications. In typical semiconductors, their electron effective mass is much lower than that of the hole. This will result in faster hot hole cooling compared to hot electron cooling. In contrast, the more "balanced" slow hot electron and hot hole cooling in perovskites will be more amenable for developing practical hot-carrier optoelectronic devices.